# Intertwined Carbon Nanotubes and Ag Nanowires Constructed by Simple Solution Blending as Sensitive and Stable Chloramphenicol Sensors

**DOI:** 10.3390/s21041220

**Published:** 2021-02-09

**Authors:** Yangguang Zhu, Xiufen Li, Yuting Xu, Lidong Wu, Aimin Yu, Guosong Lai, Qiuping Wei, Hai Chi, Nan Jiang, Li Fu, Chen Ye, Cheng-Te Lin

**Affiliations:** 1Laboratory of Environmental Biotechnology, School of Environmental and Civil Engineering, Jiangnan University, Wuxi 214122, China; zhuyangguang@nimte.ac.cn; 2Key Laboratory of Marine Materials and Related Technologies, Zhejiang Key Laboratory of Marine Materials and Protective Technologies, Ningbo Institute of Materials Technology and Engineering (NIMTE), Chinese Academy of Sciences, Ningbo 315201, China; jiangnan@nimte.ac.cn; 3College of Materials and Environmental Engineering, Hangzhou Dianzi University, Hangzhou 310018, China; xuyuting@hdu.edu.cn (Y.X.); fuli@hdu.edu.cn (L.F.); 4Key Laboratory of Control of Quality and Safety for Aquatic Products, Chinese Academy of Fishery Sciences, Beijing 100141, China; wulidong19849510@hotmail.com; 5Department of Chemistry and Biotechnology, Faculty of Science, Engineering and Technology, Swinburne University of Technology, Hawthorn, VIC 3122, Australia; aiminyu@swin.edu.au; 6Department of Chemistry, Hubei Normal University, Huangshi 435002, China; gslai@hbnu.edu.cn; 7School of Materials Science and Engineering, Central South University, Changsha 410083, China; qiupwei@csu.edu.cn; 8East China Sea Fisheries Research Institute, Chinese Academy of Fishery Sciences, Shanghai 200090, China; andychihai@126.com; 9Center of Materials Science and Optoelectronics Engineering, University of Chinese Academy of Sciences, Beijing 100049, China

**Keywords:** chloramphenicol, carbon nanotubes, silver nanowires, electrochemical sensor, voltammetric detection

## Abstract

Chloramphenicol (CAP) is a harmful compound associated with human hematopathy and neuritis, which was widely used as a broad-spectrum antibacterial agent in agriculture and aquaculture. Therefore, it is significant to detect CAP in aquatic environments. In this work, carbon nanotubes/silver nanowires (CNTs/AgNWs) composite electrodes were fabricated as the CAP sensor. Distinguished from in situ growing or chemical bonding noble metal nanomaterials on carbon, this CNTs/AgNWs composite was formed by simple solution blending. It was demonstrated that CNTs and AgNWs both contributed to the redox reaction of CAP in dynamics, and AgNWs was beneficial in thermodynamics as well. The proposed electrochemical sensor displayed a low detection limit of up to 0.08 μM and broad linear range of 0.1–100 μM for CAP. In addition, the CNTs/AgNWs electrodes exhibited good performance characteristics of repeatability and reproducibility, and proved suitable for CAP analysis in real water samples.

## 1. Introduction

Chloramphenicol (CAP) is an effective broad-spectrum antibiotic against varieties of pathogens [1,2,3], especially salmonella, vibrio cholera and escherichia coli. Therefore, CAP has been widely used for the prevention and treatment of poultry and aquatic diseases [4,5,6]. Extensively improper usage of CAP will cause a large amount of residues to accumulate in aquatic environments [7,8]. As a result, polluted aquatic ecosystem and aquaculture products pose great health hazards to humans, with dangers such as aplastic anemia, bone marrow suppression, and cardiovascular collapse [9,10,11]. Due to the serious side-effects mentioned above, the usage of CAP has been totally banned in many countries [12,13]. However, it is still being used due to its high efficiency and low cost [14,15]. Thus, it is of great necessity and emergency to develop a highly sensitive, fast, and reliable analytical method for the trace-level detection of CAP in aquatic environments.

Conventional analytical methods for CAP detection, such as chromatographic methods (high-performance liquid chromatography (HPLC), liquid chromatography-mass spectrometry (LC-MS), gas chromatography-mass spectrometry (GC-MS)), fluorescence measurement, and electrochemical detection method have been developed for decades [16,17,18]. Among them, the electrochemical detection method has been preferably accepted due to its high sensitivity, high exactitude, and time efficiency [19,20,21]. However, the method of using conventional electrochemical electrodes like glassy carbon electrodes (GCE) directly for CAP detection usually suffers from several problems including high overpotential and surface fouling. Therefore, chemical modification on GCE has become a widely used strategy, especially with various nanomaterials [22,23].

Conventionally, carbon nanomaterials (carbon nanotubes (CNTs), graphene and its derivatives), metal/metal oxide nanomaterials, and their nanocomposites have proven effective for promoting sensing performances towards CAP detection [24,25,26]. Using CNTs and metal nanoparticles (NPs) hybridized nanocomposites as the electrode is a common approach. Munawar et al., demonstrated a sensor design based on a 3D CNTs/CuNPs hybrid structure for CAP detection, achieving a detection range of 10–500 μM and a detection limit of 10 μM [27], wherein CNTs were functionalized with the amino groups to generate reaction centers for CuNPs deposition. Zhao et al., fabricated a novel composite film consisting of carboxyl acid functionalized CNTs (c-CNTs), AuNPs, and molecularly imprinted polymer (MIP), indicating good performance in sensing CAP with a detection range of 0.3–310 μM and a detection limit of 0.074 μM [28]. The nucleation of AuNPs on c-CNTs was formed in situ through the reduction of HAuCl_4_ by NaBH_4_, and the MIP film was coated onto the surface of c-CNTs/AuNPs/GCE using the thermal polymerization method. Besides CNTs, graphene is an alternative to the fabrication of carbon/metal nanocomposites and carbon/metal oxide nanocomposites [24,29,30,31,32]. Selvi et al., synthesized zinc oxide with reduced graphene oxide (ZnO/rGO) via a hydrothermal method to propose a CAP sensor with a linear range of 0.19–2857.3 μM and a detection limit of 0.13 μM [24]. Borowiec et al., developed a CAP sensor design based on an N-doped graphene/AuNPs hybrid structure, showing an excellent performance with a linear range of 2.0–80.0 μM and a low detection limit of 0.59 μM [33]. The AuNPs was deposited in situ on the surface of N-doped graphene via the reduction of HAuCl_4_ using ethylene glycol as the reducing agent. The abovementioned works were achieved through in situ deposition or chemical bonding to fabricate stable carbon/metal nanocomposites and carbon/metal oxide nanocomposites. In order to improve the convenience of electrochemical detection, it is significant to explore simple and mass-prepared methods to fabricate stable carbon/metal nanocomposites and carbon/metal oxide nanocomposites. However, few previous studies reported the fabrication of such nanocomposites for sensitive CAP detection without chemical reaction. Karthik et al., constructed a simple and rapid synthesis AuNPs decorated graphene oxide (AuNPs/GO) through physical blending to determine CAP, achieving a linear range of 1.5–2.95 μM and a detection limit of 0.25 μM [5], indicating no performance superiority when compared to the chemical synthesis method. Thus, it is necessary to further promote the simple and rapid fabrication of stable carbon/metal nanocomposites and carbon/metal oxide nanocomposites without chemical reaction towards the sensitive and selective detection of CAP. One-dimensional (1D) silver nanowires (AgNWs) possess excellent electrical conductivity and provide increased numbers of binding sites for supporting materials [34,35]. To realize the simple and rapid fabrication of composite nanomaterials, the binding of two 1D nanomaterials, such as AgNWs and CNTs, was achieved through the structural regulation of strip-shaped physical intertwining.

In this work, we attempted to fabricate CNTs/AgNWs composites through solution blending without chemical reaction. CNTs and AgNWs formed an intertwined structure in the blending process to realize a stable binding between them. The CNTs/AgNWs nanocomposites were modified on GCE as the CAP sensor. As evidenced by cyclic voltammetry (CV) and differential pulse voltammetry (DPV) tests, this proposed CNTs/AgNWs sensor displayed a low detection limit of up to 0.08 μM and a wide linear range of 0.1–100 μM for CAP. In addition, the proposed electrochemical sensor exhibited good repeatability, selectivity, and potential for practical application.

## 2. Materials and Methods

### 2.1. Chemicals

Potassium chloride (KCl), sodium chloride (NaCl), sodium sulphate (Na_2_SO_4_), dibasic sodium phosphate (Na_2_HPO_4_), potassium dihydrogen phosphate (KH_2_PO_4_), ascorbic acid, and glucose were purchased from Sinopharm Chemical Reagent Co., Ltd., Shanghai, China. Antibiotics including CAP, Erythromycin, Malachite green, Nitrofurazone, Metronidazole, and Ciprofloxacin were purchased from Shanghai Aladdin Bio-Chem Technology Co., Ltd., Shanghai, China. All of the above chemical reagents were analytical reagents and were used without further purification. CNTs (diameter: 10–20 nm, length: 10–30 μm) ethanol dispersion and AgNWs (diameter ≈ 60 nm, length ≈ 10 μm) ethanol dispersion were purchased from XFNANO Materials Technology Co., Ltd., Nanjing, China, and Nanjing JCNANO Technology Co., Ltd., Nanjing, China, respectively. Deionized Milli-Q water (18.2 MΩ/cm) was used throughout the experiments.

### 2.2. Fabrication of CNTs/AgNWs Electrodes

The CNTs/AgNWs composite was fabricated via solution blending of the CNTs ethanol dispersion and the AgNWs ethanol dispersion. In the preparation of CNTs/AgNWs composite, varied mass ratios of CNTs to AgNWs (as mCNTs/mAgNWs) were fabricated, namely 4:1, 2:1, 1:1, 1:2, and 1:4. Then they were dispersed ultrasonically for 2 h in an ice bath. Before modification, the 3 mm-diameter glassy carbon electrodes (GCEs) were polished using a 0.05 μm alumina slurry and cleaned in deionized water and ethanol by ultrasonication. Following that, the GCEs were activated via repetitive potential range scanning from −1–1 V with a scan rate of 0.1 V/s in 0.5 M H_2_SO_4_. Then the CNTs/AgNWs dispersion was uniformly dropped onto the surface of the GCEs and dried as the experimental group (CNTs/AgNWs electrodes). As controls, CNTs-modified GCEs (CNTs electrodes) and AgNWs modified-GCEs (AgNWs electrodes) were also prepared using the same method.

### 2.3. Characterizations

Field emission scanning electron microscope (FE-SEM QUANTA 250 FEG, FEI, Hillsboro, OR, USA) and energy dispersive spectroscopy (EDS)-mapping were applied to observe the morphology of the nanomaterials, including CNTs and AgNWs separately and the composite material. Raman spectroscopy (Renishaw in Via Reflex, Renishaw plc, Wotton-under-Edge, UK) and X-ray diffraction (XRD, D8 Advance, Germany) were employed for the elemental analysis and crystallite structure of the synthetic material as CNTs/AgNWs. Dynamic light scattering-zeta potential (Zeta potential, Zetasizer Nano ZS, UK) was used for the measurement of the zeta potential of nanomaterials dispersion. UV-Vis spectrophotometer (UV-Vis, Lambda 950, Waltham, MA, USA) was used for the measurement of transmittance and absorbance. All electrochemical experiments were conducted with a CHI660e electrochemical workstation (Shanghai Chenhua Co., Ltd., Shanghai, China).

### 2.4. Electrochemical Tests

The electrolyte was a phosphate buffer solution (PBS), which contained 137 mM NaCl, 102.7 mM KCl, 8.1 mM Na_2_HPO_4_, and 1.8 mM KH_2_PO_4_ (pH ≈ 7.4). The bare GCEs, CNTs electrodes, AgNWs electrodes and CNTs/AgNWs electrodes were applied as working electrodes, which were separately immersed into the PBS containing different CAP concentrations from 0.1 μM to 100 μM to compare their electrochemical properties. A saturated calomel electrode (SCE, Pt Hg(l)|Hg_2_Cl_2_(s)|KCl (saturated)) and a Pt electrode were applied as reference electrode and counter electrode, respectively. CV and DPV tests [36] were conducted to analyze the electrochemical behavior of different concentrations of CAP on the GCE modified with various materials. CV curves (six cycles) were recorded from −0.8 to 0.4 V with scan rate of 0.1 V/s, while DPV tests were conducted from −0.8 to −0.3 V with an increment step of 4 mV, amplitude of 50 mV, and pulse period of 0.5 s.

## 3. Results and Discussion

### 3.1. Characterization of CNTs/AgNWs Nanocomposite

The schematic diagram of electrochemical detection procedures of CAP on the surface of the GCE modified with the CNTs/AgNWs composite is displayed in Figure 1a. The uniform CNTs/AgNWs dispersion was prepared through blending CNTs dispersion and AgNWs dispersion with ultrasonication. The CNTs/AgNWs dispersion was dropped and dried on the polished GCEs to form CNTs/AgNWs electrodes for CAP detection. Based on previous studies (Figure 1b) [37,38,39], the irreversible Re_2_ peak at −0.69V can be interpreted as the four electron reduction of the nitro group of CAP to hydroxylamine. Meanwhile, an oxidation peak (Ox_1_) and a reduction peak (Re_1_) at 0.12 and −0.18 V, respectively, results from the two electron redox reaction between aryl hydroxylamine and a nitroso derivative. The morphologies of AgNWs, CNTs, and CNTs/AgNWs electrodes are shown in Figure 2a–c. As Ag is a rigid material, AgNWs exhibited a disorder stacking rod structure. Correspondingly, flexible CNTs intertwined with each other, forming a spaghetti-like network structure. When CNTs and AgNWs were mixed together, AgNWs were interspersed in the CNTs network randomly as shown in Figure 2c. Because of the physical constraints of the CNTs network space, AgNWs were entwined and immobilized in the nanocomposites without a chemical binder. Figure 2d is an enlarged view of Figure 2c, and the corresponding energy dispersive spectra mapping of Ag and C are shown in Figure 2e,f respectively. According to the result, the C element was well-distributed on the whole electrode and the distribution of Ag was consistent with the morphology, indicating that a minority of AgNWs were entwined by a majority of CNTs. To interpret the physical interaction that occurred between AgNWs and CNTs, the zeta potential of the CNTs, AgNWs, and CNTs/AgNWs dispersion was determined to be −24.3, −8.79, and −20.0 mV, respectively, as shown in Figure 2g. The results indicated the negative surface charges of these samples in ethanol dispersion and suggested a Van der Waals interaction between CNTs and AgNWs, which formed the physical-intertwining structure of the composite. Considering the long-term stability of electrochemical experiments about CAP detection, the storage stability of CNTs/AgNWs dispersion in ethanol at room temperature is necessary to be investigated. As arises from the results shown in Appendix A, the CNTs/AgNWs dispersion with 1:1 mCNTs/mAgNWs exhibited good storage stability without the appearance of precipitates after standing for 72 h. In order to analyze the degree of long-term stability of the dispersion, the sample was diluted 30 times to increase the transmittance for UV-Vis measurements. In Appendix A, both the transmittance and absorbance of the dispersion vary slightly with an increase of standing time (12 h), confirming a good dispersion stability of CNTs/AgNWs in ethanol at room temperature.

Raman spectra of CNTs and CNTs/AgNWs are shown in Figure 2h. Three main peaks can be seen, namely the D band (~1350 cm^−1^), G band (~1580 cm^−1^), and 2D band (~2700 cm^−1^). The formation of these three peaks can be attributed to the presence of disorder in the CNTs, the in-plane vibration of the C–C bond and the overtone of the D band, respectively [40]. As Ag has no resonance in Raman scattering, the spectrum of CNTs/AgNWs is similar to the spectrum of CNTs. Figure 2i presents the crystallographic information of AgNWs, CNTs, and CNTs/AgNWs. Five peaks at 2θ values of 38.2°, 44.4°, 64.6°, 77.6° and 81.8° can be seen in the XRD pattern of the AgNWs, which are indexed to the (111), (200), (220), (311), and (222) facets of Ag, respectively [41]. Thus, the face-centered cubic crystal structure of Ag can be confirmed. The main peaks at 2θ values of 26.0° can be observed in the XRD pattern of CNTs, which are assigned to the (002) facet [42]. As the composite, a series of Ag peaks with a weak (002) peak of C can be seen in the XRD pattern of CNTs/AgNWs. The Raman spectra and XRD patterns of the samples suggest that CNTs and AgNWs physically combined into CNTs/AgNWs in the solution blending process without chemical changes.

### 3.2. Electrochemical Behavior and Performance Optimization of CAP

To investigate the electrochemical response of CAP, CV scanning was performed on GCE in PBS with 1 mM CAP. As shown in Figure 3a, compared with the CV curve from blank PBS, there are two reduction peaks (Re_1_ and Re_2_) and one oxidation peak (Ox_1_) in the CV curve of CAP. On CNTs/AgNWs electrodes, the electrochemical responses of CAP (100 μM) and its blank control were also examined, as shown in the inset of Figure 3a. The oxidation and reduction peaks of CNTs/AgNWs electrodes occurred at 0.16 and −0.10 V in the absence of CAP, which could be attributed to the redox of Ag in aqueous solution. In this case, the Ox_1_ and Re_1_ peaks of CAP on CNTs/AgNWs electrodes overlapped with the redox peak of Ag, while only the Re_2_ peak of CAP appeared independently. Therefore, Re_2_ was specified as the characteristic peak for qualitative and quantitative analysis of the electrochemical behavior of CAP. To assess the electrochemical feasibility of various modified electrodes in 10 mM [Fe(CN)_6_]^3−/4−^, electrochemical impedance spectroscopy (EIS) was performed in 0.1 to 100 KHz on bare GCE, CNTs, AgNWs, and CNTs/AgNWs electrodes with 10 mV amplitude of the AC voltage, as shown in Figure 3b. The semicircle diameter at higher frequencies in the Nyquist diagram indicates the interfacial electron transfer resistance (R_ct_), which controls the electron transfer of [Fe(CN)_6_]^3−/4−^ on the electrode surface [33]. The R_ct_ values of GCE, CNTs, AgNWs, and CNTs/AgNWs electrodes were 532.6, 268.1, 307.3, and 53.9 Ω, respectively. The result reveals that CNTs/AgNWs nanocomposite modified GCEs greatly facilitate the electron transfer of CAP electrochemical reaction. As shown in Figure 3c, CV curves of electrochemical behaviors at a potential interval of −0.8–−0.4 V were conducted in the presence of 100 μM CAP on bare GCE, CNTs, AgNWs, and CNTs/AgNWs electrodes. The Re_2_ peak potentials of CAP on the four electrodes mentioned above were −0.69, −0.70, −0.52, and −0.53 V, respectively. Compared with the electrodes without AgNWs, the Re_2_ peak potential of AgNWs and CNTs/AgNWs electrodes both shifted positively nearly to 0.17 V, indicating that the incorporation of AgNWs facilitated the redox reaction of CAP in thermodynamics. The current intensities on CNTs and AgNWs improved to twice that of bare GCE, indicating that the modification of nanomaterials remarkably promoted the electron transfer of CAP reduction. Furthermore, Re_2_ current density reached the maximum on CNTs/AgNWs electrodes, owing to the synergistic effect of catalytically active AgNWs and the porous structure of CNTs.

To further improve the electrochemical performance of the proposed sensor, experimental parameters including the preparation of modified electrodes, electrolyte pH and volume of modified materials were optimized. To affirm the influence of the preparation of modified electrodes on the voltammetry response in PBS with 100 μM CAP, comparisons of modified electrodes were conducted with various mCNTs/mAgNWs under the equivalent total nanocomposites mass, as shown in Figure 3d. The current density increased with mCNTs/mAgNWs, varying from 4:1 to 1:1 in Figure 3e after background subtraction, which indicated that a certain amount of AgNWs added into the nanocomposite was beneficial for CAP electrochemical performance. Similarly, the current density decreased with mCNTs/mAgNWs varying from 1:1 to 1:4 when CAP adsorption of CNTs was weakened, which revealed an inferior CAP electrochemical performance with AgNWs comprising the majority of nanocomposite modified electrodes. Consequently, the equivalent mass ration of CNTs to AgNWs was the optimum for the preparation of modified electrodes, confirming the synergistic effect of the porous structure of CNTs and the catalytically active AgNWs.

The effect of pH on the electrochemical response was examined in the range from 3.0 to 11.0, as shown in Figure 3f. The peak potentials of CAP shifted positively from −0.637 to −0.463 V when electrolyte pH decreased from 11.0 to 3.0. The value of peak current density (I_pc_) reached the maximum at pH 7.0 (Figure 3g) after background subtraction, which indicated the optimal electrochemical performance of CNTs/AgNWs electrodes at pH 7.0, and was selected as the optimal pH value [43]. The linear regression equation of the reduction peak potential (E_pc_, Re_2_) versus pH is expressed as E_pc_ (V) = −0.022 pH −0.402 (R^2^ = 0.996). The slope value obtained was 43 mV pH^−1^, which is smaller than the Nernst theoretical value (59 mV pH^−1^) at 25 °C [44]. This result indicates the transfer process of the nitro group of CAP to hydroxylamine with four electrons and four protons [32]. The impact of CNTs/AgNWs volume on CV response of 100 μM CAP is depicted in Figure 3h. The current density value increased with the value varied from 1 to 5 μL in Figure 3i after background subtraction, indicating that the increased active sites on the surface of the modified electrodes was beneficial for the adsorption of CAP. At the same time, the current density value decreased as the volume varied from 7 to 9 μL, due to the formation of a thick film on the electrode hindering the electron transfer [43]. Thus, 5 μL was adopted as the optimized detection volume for CAP. Specifically, the whole electrochemical behavior and performance optimization of CAP was performed by using six individual electrodes.

Based on the electrochemical results obtained above, the electrochemical reduction mechanism of CAP would probably follow the following steps. The reduction of the nitro group of CAP to hydroxylamine with four electrons and four protons is followed by a two-electron redox reaction between aryl hydroxylamine and nitroso derivative [32]. Due to the incorporation of a silver element in the preparation of working electrodes, the redox reaction of CAP in thermodynamics was facilitated. The strip-shaped physical intertwining structure of CNTs/AgNWs composite modified electrodes exhibited good electrochemical performance towards CAP.

The electrochemical behavior of various electrodes was performed by CV in 10 mM [Fe(CN)_6_]^3−/4−^ containing 0.1 M KCl electrolyte solution at scan rates ranging from 20 to 200 mV s^−1^ (Figure 4 a–c). The observed peak currents (I_pa_ and I_pc_) both increased linearly with the square root of scan rates as shown in Figure 4d–f, indicating that CNTs, AgNWs, and CNTs/AgNWs electrodes were controlled by diffusion [45].The electroactive surface area (ESA) of various electrodes was calculated according to the Randle–Sevcik equation as I_pc_ = (2.69 × 10^5^) n^1.5^A D^0.5^ ν^0.5^ c, where I_pc_ is the reduction peak current, n is the electron numbers transferred in the reduction process, A is the ESA of the working electrodes, D is the diffusion coefficient of ferricyanide (7.6 × 10^−6^ cm^2^ s^−1^), ν is the scanning rate, and c is the concentration of ferricyanide in electrolyte (1 × 10^−5^ mol cm^−3^) [46]. The ESA of CNTs, AgNWs, and CNTs/AgNWs electrodes was estimated to be 0.089, 0.081, and 0.114 cm^2^, respectively, and the ESA of GCE can be counted approximately as its geometric area (0.071 cm^2^). The results reveal that the nanomaterial modified electrodes had a significantly enlarged ESA when compared to bare GCE, which we attribute to the high specific surface area and conductivity of nanomaterials.

### 3.3. Electrochemical Determination of CAP with Different Concentrations

The quantitative electrochemical detection of CAP on CNTs/AgNWs electrodes was conducted via CV measurements, as shown in Figure 5a. An increase of peak current density was recorded with the increasing concentration of CAP in the range from 0.1 to 100 μM. The peak was not significant in CV curves with CAP concentrations of 0.1 and 0.3 μM, and the linear range obtained by CV was 1 to 100 μM, as shown in Figure 5b after background subtraction. To further improve the sensitivity, DPV measurements were applied for CAP determination. As shown in Figure 5c, the peak currents of DPV curves increased with the increase of CAP concentration. The calibration curve of CAP was obtained on the average of peak current data as presented in Figure 5d. According to the calibration curve, the linear range of CAP detection was 0.1 to 100 μM. The linear regression equation was I_pc_ (μA) = −3.54 lg CAP (μM) −11.72 (R^2^ = 0.998). The limit of detection (LOD) of CNTs/AgNWs electrodes was calculated by using the equation LOD = 3 ζ/S [39], in which ζ is the standard deviation of the blank current, and the selectivity S is −31.05 μA μM^−1^ cm^−2^. LOD was calculated as 0.08 μM. Specifically, the quantitative electrochemical detection of CAP was conducted by CV and DPV measurements on six individual electrodes. Compared with modified electrodes prepared using various methods for CAP detection, as listed in Table 1, our CNTs/AgNWs electrodes achieved a relatively low LOD and a four-order-magnitude linear range with convenience and efficiency. Furthermore, our proposed sensor exhibited better electrochemical sensing performance towards CAP when compared to physical blending carbon/metal nanocomposites electrodes. It is noteworthy that the performance of our sensor shows comparability when compared to enzymatic method of CAP detection.

### 3.4. Repeatability, Reproducibility, Interference and Real Sample Analysis of CAP Sensors

In order to study the repeatability of the CNTs/AgNWs electrodes for CAP detection (100 μM), the Re_2_ peak currents in DPV curves were repeatedly measured 80 times on the same electrode at a potential interval of −0.65–−0.35 V. As shown in Figure 6a, reduction peak potentials of DPV curves were consistent at −0.49 V and these curves overlapped well. The inset graphic of Figure 6a depicts the enlarged view in the potential range between −0.53–−0.47 V to conveniently observe the variations of peak current data of these DPV curves. The peak currents were located at the region of −18.5–−19.9 μA among the 80 measurements and the relative standard deviation (RSD) was 2.17%. The reproducibility of CNTs/AgNWs electrodes was performed in the presence of 100 μM CAP by using six individual electrodes in DPV curves, as shown in Figure 6b, and the RSD was 2.46%. The results indicate that the CNTs/AgNWs electrodes have a good repeatability and reproducibility. The anti-interference of CNTs/AgNWs electrodes was investigated via DPV curves in PBS containing 30 μM CAP in the presence of interfering substances, such as KCl, Na_2_SO_4_, glucose, ascorbic acid, nitrofurazone, erythromycin, malachite green, and ciprofloxacin at 10-fold concentrations, and Metronidazole at 20-fold concentrations. Particularly, any compounds containing the nitro group in real waters can be reduced at similar reduction potentials to CAP, which is susceptible to interfere of CAP detection. In our research, nitrofurazone and metronidazole were chosen as nitroaromatic antibiotics to perform an interference study of the nitro group towards CAP. As shown in Figure 6c, the result suggests that these additive species did not induce obvious interference in DPV determination of CAP except metronidazole. As shown in Appendix A, the reduction peak potential of nitrofurazone shifted positively by 0.06 V towards CAP, while the peak potential and peak current of CAP remained unchanged. The results indicate that the addition of nitrofurazone did not interfere with CAP detection. In Appendix A, the reduction peak potential of metronidazole remained the same with CAP, while the peak current increased from 17.8 to 83.8 μA when a certain amount of metronidazole was added, indicating the obvious interference of metronidazole towards CAP. Thus, metronidazole should be separated before the detection of CAP in real waters containing metronidazole. In order to evaluate the practical application performance of CNTs/AgNWs electrodes as CAP sensors, recovery experiments were carried out for the determination of certain concentrations of CAP in tap water and river water samples. Before the measurement, real waters were filtered by using a cellulose membrane with 0.25 μm pore size. Then KCl was added to 0.1 M in the real samples, and the pH was adjusted to 7.0 to apply appropriate electrochemical detection of CAP. To evaluate the recovery performance of the proposed sensor, real samples were then spiked with 0.1 and 1 μM CAP, and DPV curves of CNTs/AgNWs electrodes were extracted, as shown in Appendix A. As shown in Table 2, the results demonstrate the accuracy and reliability of the fabricated sensor, indicating that the proposed CNTs/AgNWs electrodes exhibited a good potential for practical CAP determination.

## 4. Conclusions

In summary, a fast and effective approach based on simple solution blending was utilized to fabricate intertwined hybrid nanomaterials of CNTs/AgNWs. Then, CNTs/AgNWs nanomaterials were modified on GCE as electrochemical sensors to detect CAP in aqueous solutions. It was demonstrated that CNTs and AgNWs both contributed to the redox reaction of CAP in dynamics, and AgNWs was beneficial in thermodynamics as well. Owing to the stability provided by the intertwined structure, the CNTs/AgNWs electrodes exhibited good repeatability in 80 repeated experiments with the same electrode and good reproducibility by using six individual electrodes. The CNTs/AgNWs electrodes displayed a low detection limit of up to 0.08 μM and a wide linear range of 0.1–100 μM for CAP detection. In addition, the CNTs/AgNWs-based CAP sensor exhibited good selectivity and proved suitable for the analysis of real water samples. In this study we present an effective method for the electrochemical detection of CAP, and provide an inspiration for the preparation of binder-free carbon/metal nanocomposites as well. Furthermore, to achieve the widespread application of the proposed sensor to real sample analysis, the proposed sensor can be improved to increase the anti-interference performance in the presence of antibiotics containing the nitro group.

## Figures and Tables

**Figure 1 sensors-21-01220-f001:**
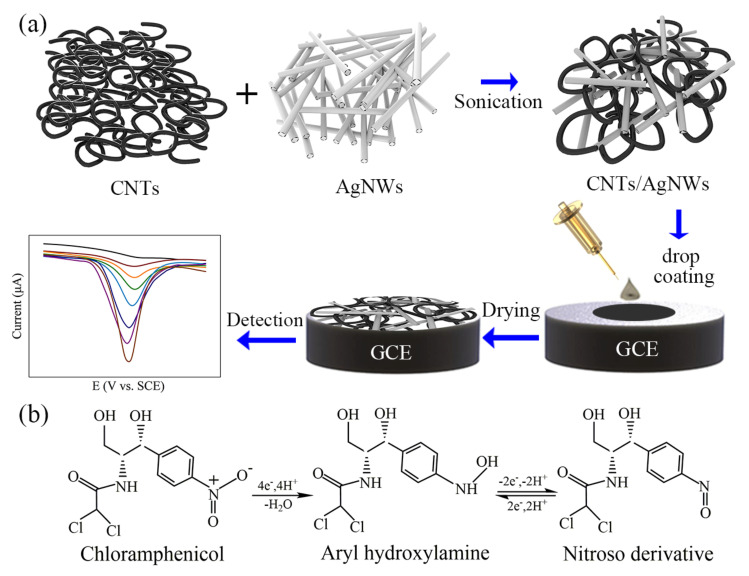
(**a**) Schematic diagram of electrochemical detection of CAP based on CNTs/AgNWs nanocomposites. (**b**) The proposed reaction scheme of redox reaction of CAP during electrochemical detection.

**Figure 2 sensors-21-01220-f002:**
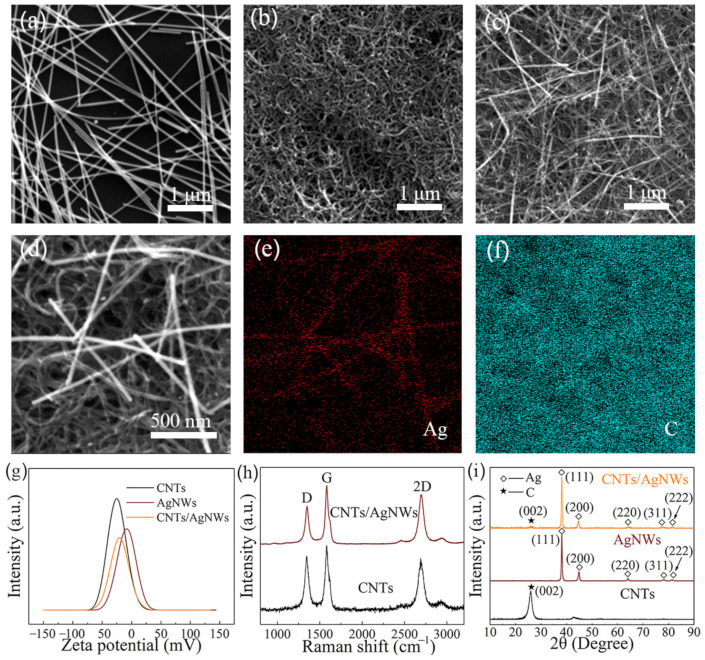
SEM images of AgNWs (**a**), CNTs, (**b**) and CNTs/AgNWs (**c**). (**d**) Regional enlarged view of the nano-intertwining structure of (**c**). The EDS mapping of element distribution of Ag (**e**) and C (**f**). (**g**) Zeta potential of CNTs, AgNWs, and CNTs/AgNWs dispersion. Raman spectra (**h**) and XRD patterns (**i**) of CNTs/AgNWs nanocomposite.

**Figure 3 sensors-21-01220-f003:**
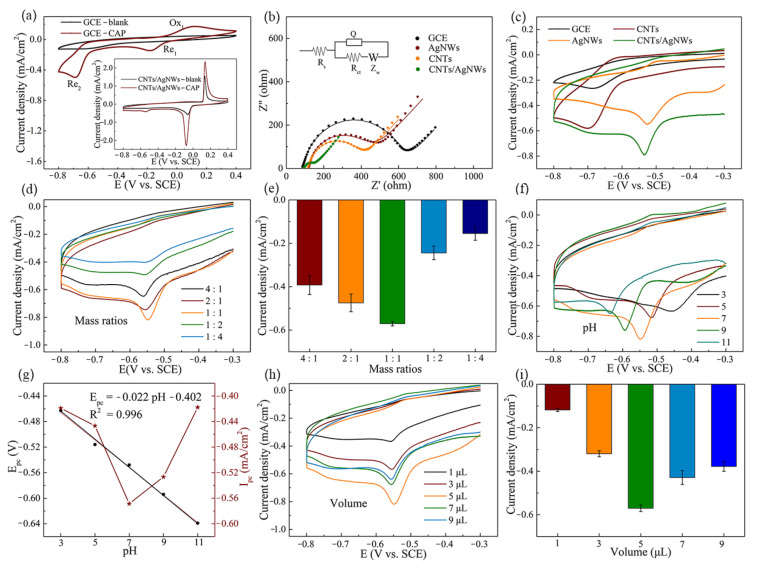
(**a**) CV of GCE with/without 1 mM CAP; the inset graphics depict CNTs/AgNWs electrodes with and without 100 μM CAP in PBS. (**b**) Impedance plots of various modified electrodes with 10 mM [Fe(CN)_6_]^3−/4−^. (**c**) Comparison of CV curves between AgNWs, CNTs, and CNTs/AgNWs electrodes with 100 μM CAP. (**d**) The preparation of modified electrodes with different mass ratios of CNTs to AgNWs and (**e**) the corresponding reduction peak current densities (background subtracted). (**f**) CV of 100 μM CAP on CNTs/AgNWs electrodes with pH. (**g**) Plots of E_pc_ and I_pc_ vs. pH (background subtracted). (**h**) CV of 100 μM CAP on CNTs/AgNWs electrodes with different volume and (**i**) the corresponding reduction peak current density (background subtracted).

**Figure 4 sensors-21-01220-f004:**
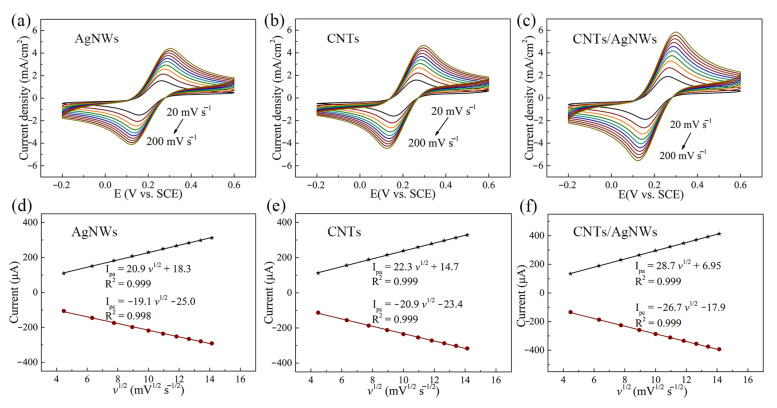
(**a**–**c**) CV of AgNWs, CNTs, and CNTs/AgNWs electrodes in 10 mM [Fe(CN)_6_]^3−/4−^ and 0.1 M KCl electrolyte solution at scan rates (*v*) from 20 to 200 mV s^−1^. (**d**–**f**) Linear plots of I_pa_/I_pc_ vs. *v*.

**Figure 5 sensors-21-01220-f005:**
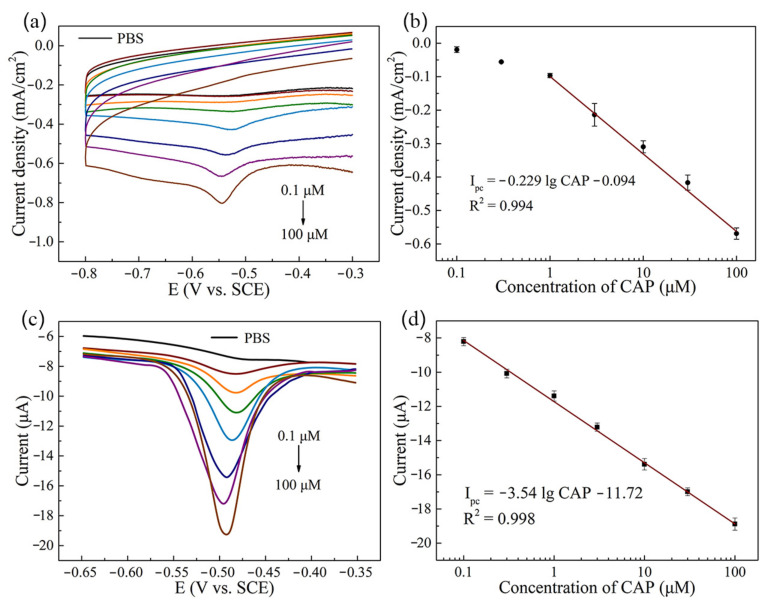
(**a**) CV curves of CNTs/AgNWs electrodes with various concentrations of CAP and (**b**) the peak current density as a function of CAP concentration (background subtracted). (**c**) DPV curves of CNTs/AgNWs electrodes for CAP detection and (**d**) the corresponding peak current versus CAP concentration.

**Figure 6 sensors-21-01220-f006:**
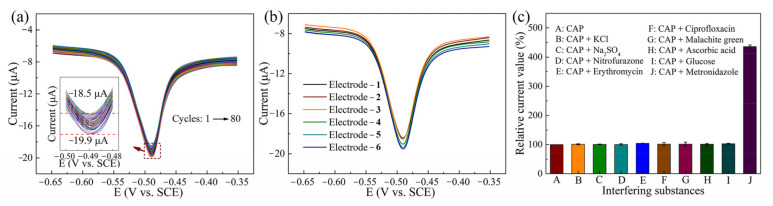
(**a**) DPV curves of CNTs/AgNWs electrodes for 100 μM CAP detection; the inset graphic depicts an enlarged view in the potential range between −0.53–−0.47 V. (**b**) The reproducibility of CNTs/AgNWs electrodes. (**c**) Good selectivity of our electrodes against interferences.

**Table 1 sensors-21-01220-t001:** Performance comparison of modified electrodes prepared using various methods for CAP detection.

Modified Electrodes	Measurements	Linear Range (μM)	LOD (μM)	Ref.
**Chemical synthesis**				
CNTs/CuNPs	CV	10–500	10	[27]
N-doped Graphene/AuNPs	LSV ^a^	2.0–80	0.59	[33]
rGO/ Co_3_O_4_	CV	1.0–2000	0.55	[36]
rGO/ ZnO	LSV ^a^	0.19–2847	0.13	[24]
rGO/ Pt-Pd nanocubes	LSV ^a^	0.20–30	0.10	[47]
rGO/PdNPs	DPV	0.05–1.0	0.05	[35]
Graphene/AgNPs	DPV	0.02–20	0.01	[32]
Poly (Eriochrome black T)	SWV ^b^	0.01–4.0	0.003	[48]
Exfoliated porous carbon	SWV ^b^	0.01–1.0	0.003	[49]
CNTs/Molecularly imprinted polymer	DPV	0.005–4.0	0.0001	[1]
**Enzymatic method**				
Alcohol dehydrogenase	Amperometry	3–5000	1	[50]
**Physical blending**				
AuNPs/GO	Amperometry	1.5–2.95	0.25	[5]
CNTs/AgNWs	DPV	0.1–100	0.08	This work

^a^ LSV: Linear Sweap voltammetry; ^b^ SWV: Square wave voltammetry.

**Table 2 sensors-21-01220-t002:** Recovery results of CAP in real water samples by using CNTs/AgNWs electrodes.

Samples	Added (μM)	Founded (μM)	RSD (%)	Recovery (%)
Tap water	0.100	0.109	1.48	109
	1.00	0.983	4.55	98.3
River water	0.100	0.111	3.03	111
	1.00	1.08	4.88	108

## Data Availability

Data sharing not applicable.

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
