# Peer review of "Intertwined Carbon Nanotubes and Ag Nanowires Constructed by Simple Solution Blending as Sensitive and Stable Chloramphenicol Sensors"

_sensors, 2021, doi:10.3390/s21041220_

Round 1

Reviewer 1 Report

1.) Revise the abstract according to the contents of the manuscript.

2.) Please elaborate the abbreviated chemical names in the line no. 83.

3.) Please include the amplitude of the AC voltage applied for the EIS recording and explain the EIS data in terms of changes in Rct at major stages of the electrode fabrication.  

4.) Show the fitting curves in addition to the experimental data points in the figure 3b.

5.) Include the electroactive surface area of the differently modified electrodes (GCE/AgNWs, GCE/CNTs and GCE/AgNWs/CNTs) under sub section 3.2.

6.) Show the square root of the scan rate vs Ipc in the inset of figure 3.i and check whether it is Ipa or Ipc in the equation added in figure 3.i.

7.) Confirm whether the redox reaction is 4 e- or 2 e- process for CAP detection through pH study.

8.) Include the formula for calculating the LOD under sub-section 3.3. Please add SD of blank or SE used for the calculation of the LOD in the text under sub-section 3.3.

9.) Add the linear regression equation in figure 4 b and 4c.

10.) Mention the concentration of CAP used for the repeatability and also add the reproducibility study with % RSD value.

11.) In real-sample analysis clearly state whether the lab water and river water was spiked with 10 µM of CAP.

12.) Add the recovery study for the real-sample analysis by doing the standard addition method for better application of the work presented in the study.

13.) Add any enzymatic methods also in the comparsion table for the determination of the CAP.

14.) Revise the conclusion by adding any limitation of the work and/or applicability of the method for determination of any other analyte's.

15.) Mention the concentration of CAP used in the selectivity study.

16.) Include whether the data is recorded on different electrodes (n =??) or with the same electrode with different measurements (n=??) in the figure 3, 4 and 5. 

17.) Please improve the mechanism of the study and response studies characterization. Improve the quality of the English in the entire manuscript.

Author Response

Dear Editor,

We are very grateful to your and the reviewers’ critical comments and thoughtful suggestions. Based on these comments, we have made a careful revision of the original manuscript (Manuscript ID: sensors-1072144). A revised manuscript has been resubmitted, of which the modified sections are marked in red. Thank you and reviewers again, who made great contribution to improve our paper. We responded point by point to the reviewers’ comments as listed below, along with a clear indication of the location in the manuscript:

Reviewer #1:

  1. Revise the abstract according to the contents of the manuscript.

Reply: Thank you for your suggestion. The abstract according to the contents of the manuscript have been revised, as mentioned in abstract in the manuscript.

  1. Please elaborate the abbreviated chemical names in the line no. 83.

Reply: Thank you for your suggestion. The abbreviated chemical names have been elaborated as “Potassium chloride (KCl), sodium chloride (NaCl), sodium sulphate (Na2SO4), dibasic sodium phosphate (Na2HPO4), potassium dihydrogen phosphate (KH2PO4), ascorbic acid and glucose..”. A description about this conclusion has been added in the manuscript in Section 2.1 in Materials and Methods (Line 1 – 2, Page 2).

  1. Please include the amplitude of the AC voltage applied for the EIS recording and explain the EIS data in terms of changes in Rct at major stages of the electrode fabrication.  

Reply: Thank you for your suggestion. The amplitude of the AC voltage applied for the EIS recording is included as 10 mV. According to the fitting curves in addition to the experimental data points in Figure R1, the Rct value of GCE, CNTs, AgNWs and CNTs/AgNWs electrodes were 532.6 Ω, 268.1 Ω, 307.3 Ω and 53.9 Ω, respectively. The result revealed that CNTs/AgNWs nanocomposite modified GCE greatly facilitates the electron transfer of CAP electrochemical reaction. A description about this conclusion has been added in the manuscript in Section 3.2 in Results and Discussion (Line 8 – 13, Page 5).

Figure R1. Impedance plots of various modified electrodes with 10 mM [Fe(CN)6]3-/4-.

  1. Show the fitting curves in addition to the experimental data points in the figure 3b.

Reply: Thank you for your suggestion. The fitting curves in addition to the experimental data points have been added in Figure R1.

  1. Include the electroactive surface area of the differently modified electrodes (GCE/AgNWs, GCE/CNTs and GCE/AgNWs/CNTs) under sub section 3.2.

Reply: Thank you for your suggestion. The electroactive surface area of the differently modified electrodes (AgNWs, CNTs and AgNWs/CNTs) has been included under sub section 3.2. As shown in Figure R2, the electrochemical behavior of various electrodes was performed in CV voltammograms by using 10 mM [Fe(CN)6]3-/4- containing 0.1 M KCl as redox solution at different scan rates ranging from 20 to 200 mV s-1. The electroactive surface area (ESA) of AgNWs, CNTs and AgNWs/CNTs electrodes were calculated according to the Randle–Sevcik equation, which were 0.089 cm2, 0.081 cm2 and 0.114 cm2, respectively. A description about this conclusion has been added in the manuscript in Section 3.2 in Results and Discussion (Paragraph 4 , Page 7-8).

Figure R2. (a-c) CV of AgNWs, CNTs and CNTs/AgNWs electrodes in 10 mM [Fe(CN)6]3-/4- and 0.1 M KCl electrolyte solution at scan rates (v) from 20 to 200 mV s-1. (d-f) Linear plots of Ipa/Ipc vs. v.

  1. Show the square root of the scan rate vs Ipcin the inset of figure 3.i and check whether it is Ipa or Ipc in the equation added in figure 3i.

Reply: Thank you for your suggestion. In order to obtain the electroactive surface area of the differently modified electrodes (AgNWs, CNTs and AgNWs/CNTs), the electrochemical behavior of various electrodes was performed in CV voltammograms by using 10 mM [Fe(CN)6]3-/4- containing 0.1 mM KCl as redox solution. The linear plot of observed peak current (Ipa and Ipc) versus square root of scan rates is shown in Figure R2. A description about this conclusion has been added in the manuscript in Section 3.2 in Results and Discussion (Paragraph 4 , Page 7-8).

  1. Confirm whether the redox reaction is 4 e-or 2 e- process for CAP detection through pH study.

Reply: Thank you for raising this question. The reduction peak potential (Epc, Re2) shifted positively from -0.637 V to -0.463 V with electrolyte pH decreased from 11.0 to 3.0. The linear regression equation of Epc versus pH is expressed as Epc (V) = -0.043 pH -0.423 (R2 = 0.996). The slope value obtained was 43 mV pH-1, which is smaller than the Nernst theoretical value (59 mV pH-1) at 25 °C [1]. The result indicated the equal number of protons and electrons involved in the electrochemical reduction of CAP. Based on previous studies [2,3], the irreversible Re2 peak at -0.69V can be interpreted as the four electron reduction of nitro group of CAP to hydroxylamine. So we can confirm the transfer process of nitro group of CAP to hydroxylamine with four electrons and four protons. A description about this conclusion has been added in the manuscript in Section 3.2 (paragraph 3, Line 5 – 8, Page 7).

  1. Include the formula for calculating the LOD under sub-section 3.3. Please add SD of blank or SE used for the calculation of the LOD in the text under sub-section 3.3.

Reply: Thank you for your suggestion. The formula for calculating the LOD under sub-section 3.3 have been included, and SD of blank current for the calculation of LOD also have been included in the text. The linear regression equation was Ipc (μA) = - 3.54 lg CAP (μM) - 11.72 (R2 = 0.998). The low detection limit (LOD) of CNTs/AgNWs electrodes was calculated by using the equation LOD = 3 ζ/S [4], in which ζ is the standard deviation of the blank current calculated as 0.094 μA, and the selectivity S is -31.05 μA μM-1 cm-2. LOD was calculated as 0.08 μM. A description about this conclusion has been added in the manuscript in Section 3.3 (Line 8 – 13, Page 8).

  1. Add the linear regression equation in figure 4 b and 4c.

Reply: Thank you for your suggestion. As shown in Figure 5b and 5c in the manuscript, the linear regression equation have been included. The linear regression equation of CV curves was Ipc (μA) = - 0.229 lg CAP (μM) – 0.094 (R2 = 0.994). And the linear regression equation of DPV curves was Ipc (μA) = - 3.54 lg CAP (μM) - 11.72 (R2 = 0.998).

  1. Mention the concentration of CAP used for the repeatability and also add the reproducibility study with % RSD value.

Reply: Thank you for your suggestion. The concentration of CAP used for the repeatability have been mentioned as 100 μM in Figure 6a caption. The reproducibility of CNTs/AgNWs electrodes had been performed by using six individual electrodes in DPV curves as shown in Figure R3, and the RSD was 2.46%. A description about this conclusion has been added in the manuscript in Section 3.4 (Line 6 – 9, Page 9).

Figure R3. The reproducibility of CNTs/AgNWs electrodes.

  1. In real-sample analysis clearly state whether the lab water and river water was spiked with 10 µM of CAP.

Reply: Thank you for your suggestion. The lab water and river water spiked with 1 µM and 10 µM of CAP have been clearly stated in real-sample analysis. A description about this conclusion has been added in the manuscript in Section 3.4 (Line 18, Page 9).

  1. Add the recovery study for the real-sample analysis by doing the standard addition method for better application of the work presented in the study.

Reply: Thank you for your suggestion. In order to evaluate the practical application performance of CNTs/AgNWs electrodes as CAP sensors, recovery experiments have been performed for the determination of certain concentrations of CAP in real water samples, as shown in Table R1 and Figure R4. A description about this conclusion has been added in the manuscript in Section 3.4 (Line 14 – 20, Page 9).

Figure R4. DPV curves of CNTs/AgNWs electrodes for real sample analysis.

Table R1. Recovery results of CAP in real water samples by using CNTs/AgNWs electrodes.

Samples

Added (μM)

Founded (μM)

RSD (%)

Recovery (%)

Tap water

1

1.073

4.58

107.3

10

9.77

4.47

97.7

River water

1

1.097

4.76

109.7

10

10.700

3.55

107.0

  1. Add any enzymatic methods also in the comparison table for the determination of the CAP.

Reply: Thank you for your suggestion. An enzymatic method used an enzymatic full cell device for the determination of chloramphenicol is added in Table 1 in the manuscript, as also shown below (Table R1).

Table 1. Performance comparison of various electrodes prepared by various methods for CAP detection.

Modified electrodes

Measurements

Linear range (μM)

LOD (μM)

ref

Chemical synthesis

CNTs/CuNPs

CV

10 – 500

10

[27]

N-doped Graphene/AuNPs

LSV

2 – 80

0.59

[33]

rGO/ Co3O4

CV

1 – 2000

0.55

[36]

rGO/ ZnO

LSV

0.19 – 2847.3

0.13

[24]

rGO/ Pt-Pd nanocubes

LSV

0.2 – 30

0.1

[47]

rGO/PdNPs

DPV

0.05 – 1

0.05

[35]

Enzymatic method

Alcohol dehydrogenase

Amperometry

3 – 5000

1

[48]

Physical blending

AuNPs/GO

Amperometry

1.5 – 2.95

0.25

[5]

CNTs/AgNWs

DPV

0.1 – 100

0.08

This work

  1. Revise the conclusion by adding any limitation of the work and/or applicability of the method for determination of any other analytes'.

Reply: Thank you for your suggestion. Any compounds containing nitro group in real waters can be reduced at similar reduction potential as CAP, which is susceptible to interfere CAP detection. Thus, to prospect the widespread application of the proposed sensor to the real-sample analysis, the anti-interference study of proposed sensor in the presence of any compounds containing nitro group should be further discussed in the future. The conclusion by adding any limitation of the work have been revised and added in the manuscript in Section 4 in Conclusions (Line 10 – 13, Page 10).

  1. Mention the concentration of CAP used in the selectivity study.

Reply: Thank you for your suggestion. The concentration of 30 μM CAP used in the selectivity study have been mentioned in the manuscript in Section 3.4 (Line 10, Page 9).

  1. Include whether the data is recorded on different electrodes (n =??) or with the same electrode with different measurements (n=??) in the figure 3, 4 and 5. 

Reply: Thank you for raising this question. The whole electrochemical behavior and performance optimization of CAP was performed by using six individual electrodes in Figure 3-5. The quantitative electrochemical detection of CAP was conducted by CV and DPV measurements on six individual electrodes in Figure 5. To study the repeatability of the CNTs/AgNWs electrodes for CAP detection, the Re2 peak current in DPV curves were repeatedly measured by 80 measurements on the same electrode in Figure 6a. The reproducibility of CNTs/AgNWs electrodes was performed by using six individual electrodes in DPV curves as shown in Figure 6b. The interference study data in DPV curves was recorded by using six individual electrodes in Figure 6c.

  1. Please improve the mechanism of the study and response studies characterization. Improve the quality of the English in the entire manuscript.

Reply: Thank you for your suggestion. The reduction of nitro group of CAP to hydroxylamine with four electrons and four protons, followed by two electron redox reaction between aryl hydroxylamine and nitroso derivative [32]. Due to the corporation of silver element in the preparation of working electrodes, the redox reaction of CAP in thermodynamics was facilitated. Consequently, CNTs/AgNWs electrodes exhibited good electrochemical performance towards CAP, owing to the synergistic effect of porous structure from CNTs and catalytically active AgNWs. The electrochemical reduction mechanism of CAP and the response characterization was extracted and concluded, as mentioned in the manuscript in Section 3.2 in paragraph 3, Page 7).

References:

[1] Mani, V.; Dinesh, B.; Chen, S.M.; Saraswathi, R. Direct electrochemistry of myoglobin at reduced graphene oxide-multiwalled carbon nanotubes-platinum nanoparticles nanocomposite and biosensing towards hydrogen peroxide and nitrite. Biosens. Bioelectron, 2014, 53, 420-427.

[2] Yia, W.W.; Lia, Z.Q.; Dong, C.; Li, H.W.; Li, J.F. Electrochemical detection of chloramphenicol using palladium nanoparticles decorated reduced graphene oxide. Microchem. J, 2019, 148, 774-783.

[3] Yadav, M.; Ganesan, V.; Gupta, R.; Yadav, D.K.; Sonkar, P.K. Cobalt oxide nanocrystals anchored on graphene sheets for electrochemical determination of chloramphenicol. Microchem. J, 2019, 146, 881-887.

[4] Viliana, A.T.E.; Ohb, S.Y.; Rethinasabapathy, M.; Umapathi, R.; Hwang, S.K.; Oh, C.W.; Park, B.; Huh, Y.S.; Han, Y.K. Improved conductivity of flower-like MnWO4 on defect engineered graphitic carbon nitride as an efficient electrocatalyst for ultrasensitive sensing of chloramphenicol. J. Hazard. Mater, 2020, 399, 122868.

[5] Zhai, H.Y.; Liang, Z.X.; Chen, Z.G.; Wang, H.H.; Liu, Z.P.; Su, Z.H.; Zhou, Q. Simultaneous detection of metronidazole and chloramphenicol by differential pulse stripping voltammetry using a silver nanoparticles/ sulfonate functionalized graphene modified glassy carbon electrode. Electrochim. Acta, 2015, 171, 105-113.

We appreciate for Editor/Reviewers’ warm work earnestly, and hope that the correction will meet with approval. The manuscript has been overall checked, and the changes marked in red font one by one. We hope that these revisions are sufficient to make our manuscript acceptable for publication in Sensors. If you have any question about this paper, please do not hesitate to contact me.

Yours sincerely,

Cheng-Te Lin

Ningbo Institute of Material Technology & Engineering, Chinese Academy of Sciences

Reviewer 2 Report

The detection of Chloramphenicol (CAP) using a composite electrode made from carbon nanotubes and silver wire by solution blending approach is reported by Zhu et al. This work has certain merits like preparation of a conducting composite by solution blending , simple and scalable approach and the performance reported being comparbale to or better than the chemical reduction based synthesis methods. But, there is no clarity in the materials selection, experimental design and writing, and as a result it is very difficult to follow this work. The authors have compared the performance of CNTs/AgNWs prepared by solution blending method to other metal-carbon composites prepared by chemical reduction assisted synthesis methods. Though the materials' performance seems better than the rest, they should prepare the CNTs/AgNWs composite by chemical reduction method and compare it with the one that was prepared by solution blending. In the introduction, authors did not explain why Ag NWs were selected for this study and what makes Ag NWs superior to other metal nanowires? The sentence "The nanocomposite consisted of 0.5 mg/mL AgNWs dispersion and 2, 1, 0.5, 0.25, 0.125 mg/mL CNTs dispersion respectively." in section 2.2 is not clear. The authors should provide the mass ratios of CNT to AgNWs instead of mentioning their individual masses in mg/mL. It is important to metion the potential range and the scan rate at which the GCE was activated in 0.5 M H2SO4. The sentence in section 2.4 is not clear..Parameters in CV tests were recorded from -0.8 – 0.4 V with scan rate of 0.1 112 V/s in 12 segments.. Please note that one can set the parameters but cannot record them. It is better to say cycles instead of segments. I don't think the peak seen at 43.6°(100) for CNT is worthy to note and it is not seen in the composite as well. Does any sort of physical interaction occurred between Ag NWs and CNTs? The authors should determine the surface charges of CNT, AgNWs and CNT/AgNWs by Zeta potential measurements and should explain the physical interactions based on the results. How stable is the CNTs/AgNWs dispersion at room temperature. It is important to evaluate the storage stability of the CNTs/AgNWs dispersion. In Fig. 3C, Why the background current shifted for all the materials, including the composite? It is seen from the CV curves that the reduction current for the composite in particular shifted significantly, comparing to bare GCE, CNT and AgNW GCEs. The authors should explain this unique and abnormal electrochemical behavior. The reduction current shifting to higher values with increasing material loadings is also seen in Fig. 3g , which is not clear. The reduction currents in Fig. 4a are also shifting with CAP concentration, which is not explained as well. In this case, the peak currents measured to preare the calibration curves can not be considered as the real peak currents for CAP as there is a huge contribution from the background current. It is mandatory to perform background subtraction to eliminate the background current contribution. The effect of mass ratio of CNTs to AgNWs and vise versa towads CAP reseponse is not clear as the currents decreased with an increase in mass ratios. The current obtained for the 4:1 mass ratio is lesser than 1:1, but it is mentioned in the text that the current increased from 4:1 to 1:1, which is not the case and it is a bit confusing. Overall, the language and tense used throughout the manuscript is poor and there are lots of grammatical mistakes.

Author Response

Dear Editor,

We are very grateful to your and the reviewers’ critical comments and thoughtful suggestions. Based on these comments, we have made a careful revision of the original manuscript (Manuscript ID: sensors-1072144). A revised manuscript has been resubmitted, of which the modified sections are marked in red. Thank you and reviewers again, who made great contribution to improve our paper. We responded point by point to the reviewers’ comments as listed below, along with a clear indication of the location in the manuscript:

Reviewer #2:

  1. The detection of Chloramphenicol (CAP) using a composite electrode made from carbon nanotubes and silver wire by solution blending approach is reported by Zhu et al. This work has certain merits like preparation of a conducting composite by solution blending , simple and scalable approach and the performance reported being comparable to or better than the chemical reduction based synthesis methods. But, there is no clarity in the materials selection, experimental design and writing, and as a result it is very difficult to follow this work.

Reply: Thank you for your comments. The contents in the materials selection, experimental design and writing have been revised in entirety and an expectation of the suitability of following our work can be made.

  1. The authors have compared the performance of CNTs/AgNWs prepared by solution blending method to other metal-carbon composites prepared by chemical reduction assisted synthesis methods. Though the materials' performance seems better than the rest, they should prepare the CNTs/AgNWs composite by chemical reduction method and compare it with the one that was prepared by solution blending.

Reply: Thank you for making this suggestion. Although the decoration of the CNT surface with AgNPs using chemical synthesis routes has been extensively investigated [1-3], however, to the best of our knowledge, so far the preparation of CNTs/AgNWs composites by chemical reduction method has not reported yet, and the usual way for the fabrication of CNTs/AgNWs composites is still based on the physical mixing [4,5]. Due to the short deadline for revision, we feel sorry for not being able to complete this experimental comparison right now, but the future work will be carried out based on your valuable comments.

  1. In the introduction, authors did not explain why Ag NWs were selected for this study and what makes Ag NWs superior to other metal nanowires?

Reply: We acknowledge your question. Noble metal nanomaterials, such as Au, Ag and Pt, usually exhibit high electrocatalytic activity towards target compounds [6]. Particularly, the improvement of sensor performance using AgNPs, AgNWs, or their composites has been widely studied, due to their excellent electrical conductivity, increased binding sites for base materials, and the acceleration of electron transfer during electrochemical detection [7]. In addition, the cost of using Ag is obviously lower than using Au and Pt, showing superiority to promote the potential practical applications. A description has been added in the manuscript in the Introduction section (Line 26 – 29, Page 2).

  1. The sentence "The nanocomposite consisted of 0.5 mg/mL AgNWs dispersion and 2, 1, 0.5, 0.25, 0.125 mg/mL CNTs dispersion respectively." in section 2.2 is not clear. The authors should provide the mass ratios of CNT to AgNWs instead of mentioning their individual masses in mg/mL.

Reply: Thank you for your suggestion. In the preparation of CNTs/AgNWs composite, varied mass ratios of CNTs to AgNWs (namely as mCNTs/mAgNWs) were fabricated as 4 : 1, 2 : 1, 1 : 1, 1 : 2 and 1 : 4, respectively. This description have been mentioned in the manuscript in Section 3.2 (Line 2-3, Page 3).

  1. It is important to mention the potential range and the scan rate at which the GCE was activated in 0.5 M H2SO4.

Reply: Thank you for your suggestion. The potential range and scan rate at which the GCE was activated in 0.5 M H2SO4 have been mentioned. GCE was activated by repetitive potential range scanning from -1 – 1 V with scan rate of 0.1 V/s in 0.5 M H2SO4. This description have been mentioned in the manuscript in Section 2.2 (Line 5-6, Page 3).

  1. The sentence in section 2.4 is not clear. Parameters in CV tests were recorded from -0.8 – 0.4 V with scan rate of 0.1 V/s in 12 segments. Please note that one can set the parameters but cannot record them. It is better to say cycles instead of segments.

Reply: Thank you for your suggestion. “segments” have been revised to “cycles”. Parameters in CV tests were recorded from -0.8 – 0.4 V with scan rate of 0.1 V/s in 12 cycles, as mentioned in the manuscript in Section 2.4 (Line 8, Page 3).

  1. I don't think the peak seen at 43.6°(100) for CNT is worthy to note and it is not seen in the composite as well.

Reply: Thank you for your suggestion. The description about “(100) facet of C” and “Due to the large intensity of Ag peaks, the (100) peak of C is too weak to be observed” have been removed of the contents. Also the label of the (100) facet of C in Figure 2h have been removed.

  1. Does any sort of physical interaction occurred between AgNWs and CNTs? The authors should determine the surface charges of CNT, AgNWs and CNT/AgNWs by Zeta potential measurements and should explain the physical interactions based on the results.

Reply: We acknowledge your question. As shown in Figure R1, the Zeta potential of CNTs, AgNWs, and CNT/AgNWs dispersions in ethanol media was determined to be -24.3 mV, -8.8 mV, and -20.0 mV, respectively, indicating that these samples has a negative surface charge in water. According to the results of Zeta potential measurements, we suggest that there is the Van der Waals interaction between AgNWs and CNTs in our composite system.

Figure R1. Zeta potential of CNTs, AgNWs and CNTs/AgNWs dispersion.

  1. How stable is the CNTs/AgNWs dispersion at room temperature. It is important to evaluate the storage stability of the CNTs/AgNWs dispersion.

Reply: According to your comments, the storage stability of CNTs/AgNWs dispersion in ethanol at room temperature is investigated. As the results shown in Figure R2a, the ethanol dispersion composed of 0.5 mg L-1 CNTs and 0.5 mg L-1 AgNWs exhibits good storage stability without the appearance of precipitates after standing for 72 h. In order to analyze the degree of long-term stability of CNTs/AgNWs dispersion, the sample was diluted 30 times to increase the transmittance for UV-Vis measurements. In Figure R2b, both the transmittance and absorbance of the dispersion vary slightly with an increase of standing time (12 h), confirming good dispersion stability of CNTs/AgNWs in ethanol at room temperature.

Figure R2. (a) The storage stability test of CNTs/AgNWs dispersion in ethanol at room temperature. (b) The transmittance and absorbance of the dispersion as a function of standing time.

  1. In Fig. 3C, Why the background current shifted for all the materials, including the composite? It is seen from the CV curves that the reduction current for the composite in particular shifted significantly, comparing to bare GCE, CNT and AgNWs GCEs. The authors should explain this unique and abnormal electrochemical behavior. The reduction current shifting to higher values with increasing material loadings is also seen in Fig. 3g , which is not clear. The reduction currents in Fig. 4a are also shifting with CAP concentration, which is not explained as well. In this case, the peak currents measured to prepare the calibration curves can not be considered as the real peak currents for CAP as there is a huge contribution from the background current. It is mandatory to perform background subtraction to eliminate the background current contribution.

Reply: Thank you for raising these professional and academic questions.

(1) When CV performs with nanomaterials modified GCE, double layer capacitance forms on the surface of the GCE modified in the electrolyte. The increased electrochemical surface area (ESA) of electrodes enlarge the double layer capacitance, resulting lager background currents. As the synergistic effect of porous structure from CNTs and catalytically active AgNWs, the maximum ESA of CNTs/AgNWs electrodes triggered larger background current. As for Fig. 3g, excess dropping amount of nanomaterials cause the agglomeration and reduce the ESA. Thus the optimal dropping amount of nanomaterials increased the ESA the most and the background current was the enlarged to an extent.

(2) We accept the opinion that the peak currents measured to prepare the calibration curves can not be considered as the real peak currents for CAP. Due to the short deadline for revision, we feel very sorry for not being able to conduct the corresponding experiments to perfect the CV curves involved in the questions mentioned above. But the values of peak currents have been updated to prepare the calibration curves through background subtraction to eliminate the background current contribution. The corresponding data concerning the qualitative analysis of CV curves have been updated, as shown in Figure R3. According to the comparison of the original CV curves and the corresponding current values before and after updated, it’s obvious that the conclusions obtained from the experiments remain consistent.

Figure R3. (a) The original reduction peak currents corresponding to different mass ratios of CNTs to AgNWs. (b) The corresponding reduction peak currents density (before background subtraction), and (c) after background subtraction. (d) CV curves vs CAP concentration. (e) Linear plots of the peak current density of CV curves Ipc vs. CAP concentration (before background subtraction), and (f) after background subtraction.

  1. The effect of mass ratio of CNTs to AgNWs and vise versa towards CAP response is not clear as the currents decreased with an increase in mass ratios. The current obtained for the 4:1 mass ratio is lesser than 1:1, but it is mentioned in the text that the current increased from 4:1 to 1:1, which is not the case and it is a bit confusing.

Reply: Thank you for raising this question. As mentioned in the text, the mass ratio of CNTs to AgNWs is classified to 4 : 1, 2 : 1, 1 : 1, 1 : 2 and 1 : 4, respectively. The increase in mass ratios of CNTs to AgNWs means that the mass of CNTs is increased. As shown in Figure 3e, the current increased from 1:4 to 1:1, while decreased from 1:1 to 1:4. In other words, The current obtained for the 4:1 mass ratio is lesser than 1:1, which means the current increased from 4:1 to 1:1.

  • Overall, the language and tense used throughout the manuscript is poor and there are lots of grammatical mistakes. 

Reply: Thank you for your suggestion. The language and tense used throughout the manuscript have been improved, and grammatical mistakes have been corrected.

References:

[1] Liu, Y.; Tang, J.; Chen, X.; Chen, W.; Pang, G.; Xin, J. A wet-chemical route for the decoration of CNTs with silver nanoparticles. Carbon, 2006, 44, 381-383.

[2] Ko, W-Y.; Huang, L-T.; Lin, K-J. Green technique solvent-free fabrication of silver nanoparticle–carbon nanotube flexible films for wearable sensors. Sensor. Actuat. A-Phys, 2021, 317, 112437.

[3] Hou, X.; Wang, L.; Wu, R. In Situ Synthesis of Highly Dispersed Silver Nanoparticles on Multi-walled Carbon. Bull. Korea. Chem. Soc, 2011, 32, 2527.

[4] Oluwalowo, A.; Nguyen, N.; Zhang, S.; Park, J.; Liang, R. Electrical and thermal conductivity improvement of carbon nanotube and silver composites. Carbon, 2019, 146, 224-231.

[5] Xu, Y.; Yuan, Y.; Fan, X.; Cui, M.; Xiao, J.; Du, J.; Pan, Z.; Feng, G.; Lv, B.; Song, C.; Wang, t. Silver nanowire-carbon nanotube/coal-based carbon composite membrane with fascinating antimicrobial ability and antibiofouling under electrochemical assistance. J. Water. Process. Eng, 2020, 38 ,101617.

[6] Zhang, C.; Govindaraju, S.; Giribabu, K.; Huh, Y.S.; Yun, K. AgNWs-PANI nanocomposite based electrochemical sensor for detection of 4-nitrophenol. Sensor. Actuat. B-Chem, 2017, 252, 616-623. 

[7] Schoen, D.; Schoen, A.; Hu, L.; Kim, H.; Heilshorn, S.; Cui, Y. High Speed Water Sterilization Using One-Dimensional Nanostructures. Nano Letter, 2010, 10, 3628-3632.

We appreciate for Editor/Reviewers’ warm work earnestly, and hope that the correction will meet with approval. The manuscript has been overall checked, and the changes marked in red font one by one. We hope that these revisions are sufficient to make our manuscript acceptable for publication in Sensors. If you have any question about this paper, please do not hesitate to contact me.

Yours sincerely,

Cheng-Te Lin

Ningbo Institute of Material Technology & Engineering, Chinese Academy of Sciences

Reviewer 3 Report

The article presents a method for the determination of the antibiotic chloramphenicol (CAP) based on its electrochemical reduction at the surface of a glassy electrode modified with silver nanowires and carbon nanotubes. The manuscript is clearly written. The originality of the article resides in the simple method of combining the nanomaterials for obtaining an enhanced response to CAP. The sensor has good analytical characteristics, similar to other sensors employing nanomaterials but operates at relatively high negative potential being prone to interferences. The interference study and the attempt to illustrate the sensor’s potential for the analysis of river water are weak. In fact, the data presented don’t demonstrate the sensor’s reproducibility and accuracy required for practical applications. Based on all this, the manuscript is not recommended for publication in Sensors in its current form.

Specific comments and suggestions:

Figure 3b caption correct the spelling in “impendance”

Lines 197-198: discuss the variation of the peak potential with pH related to the expected number of protons and electrons involved in the electrochemical reduction of CAP

Lines 217-218: please give the criteria for establishing the detection limit. Is the lower limit of the calibration range the same as the detection limit? Why?

-What is the reproducibility of the sensor?

Lines 245-246 Please specify the concentrations of CAP and interfering compounds used.

 Interference study and figure 5b: the interference study is very limited. If the envisaged application is the detection of CAP in river water, the authors should consider more possibly interfering compounds specific to river water. Also, any compounds with a nitro group that can be reduced at similar potential as CAP is susceptible to interfere. See for example Zhai et al, 2015, doi: 10.1016/j.electacta.2015.03.140

The accuracy of the sensor should be investigated with a set of spiked river samples, if indeed river water is the target sample matrix. The spiking procedure and any dilution of the spiked sample should be clearly explained. The spiking amounts should be consistent with any maximum residue limits or with typical concentrations of CAP reported for river water.

Lines 250-252: the conclusion is contradicted by the data in figure 5c: “Apparently, as shown in Figure 5c, no obvious differences  in the potential and current of the characteristic peak were existed between these two samples, indicating that the proposed CNTs/AgNWs electrodes exhibited a good potential for practical CAP determination”. Actually , in figure 5c, the peak height for CAP reduction is quite different in river water than in purified water (“Laboratorial water”)  Please investigate the effect of sample matrix and correct the sample preparation to enable accurate measurements.

Author Response

Dear Editor,

We are very grateful to your and the reviewers’ critical comments and thoughtful suggestions. Based on these comments, we have made a careful revision of the original manuscript (Manuscript ID: sensors-1072144). A revised manuscript has been resubmitted, of which the modified sections are marked in red. Thank you and reviewers again, who made great contribution to improve our paper. We responded point by point to the reviewers’ comments as listed below, along with a clear indication of the location in the manuscript:

Reviewer #3:

  1. Figure 3b caption correct the spelling in “impendance”.

Reply: Thank you for your suggestion. The spelling in “impendance” have been corrected as “impedance”, as mentioned in the manuscript in Figure 3b caption.

  1. Lines 197-198: discuss the variation of the peak potential with pH related to the expected number of protons and electrons involved in the electrochemical reduction of CAP?

Reply: Thank you for raising this question. The reduction peak potential (Epc, Re2) shifted positively from -0.637 V to -0.463 V with electrolyte pH decreased from 11.0 to 3.0. The linear regression equation of Epc versus pH is expressed as Epc (V) = -0.043 pH -0.423 (R2 = 0.996). The slope value obtained was 43 mV pH-1,which is smaller than the Nernst theoretical value (59 mV pH-1) at 25 °C [1]. The result indicated the equal number of protons and electrons involved in the electrochemical reduction of CAP. Based on previous studies [2,3], the Re2 peak at -0.69V can be interpreted as the four electron reduction of nitro group of CAP to hydroxylamine. So we can confirm the transfer process of nitro group of CAP to hydroxylamine with four electrons and four protons. A description about this conclusion has been added in the manuscript in Section 3.2 (Paragraph3, Line 5-8, Page 7).

  1. Lines 217-218: please give the criteria for establishing the detection limit. Is the lower limit of the calibration range the same as the detection limit? Why?

Reply: Thank you for your suggestion. The criteria for establishing the detection limit have been given. The formula for calculating the low detection limit (LOD) of CNTs/AgNWs electrodes was evaluated by using the equation LOD = 3 ζ/S [4], in which ζ is the standard deviation of the blank current calculated as 0.094 μA, and the selectivity S is -31.05 μA μM-1 cm-2. The linear regression equation was Ipc (μA) = - 3.54 lg CAP (μM) - 11.72 (R2 = 0.998), and LOD was calculated as 0.08 μM. Thus, LOD (0.08 μM) obtained through equation calculation is lower than the lower limit of the calibration range (0.1 μM). A description about this conclusion has been added in the manuscript in Section 3.3 (Line 7 – 11, Page 8).

  1. What is the reproducibility of the sensor?

Reply: Thank you for raising this question. In our work, the reproducibility of CNTs/AgNWs electrodes had been performed by using six individual electrodes in DPV curves as shown in Figure R1, and the RSD was 2.46%, indicating a good reproducibility of the sensor.

Figure R1. The reproducibility of CNTs/AgNWs electrodes.

  1. Lines 245-246 Please specify the concentrations of CAP and interfering compounds used.

Reply: Thank you for your suggestion. The concentrations of CAP and interfering compounds have been specified in the manuscript. 30 μM CAP was chosen for electrochemical detection in the presence of interfering substances at 10 - fold concentrations, as mentioned in the manuscript in Section 3.4 (Line 9 – 10, Page 9).

  1. Interference study and figure 5b: the interference study is very limited. If the investigated application is the detection of CAP in river water, the authors should consider more possibly interfering compounds specific to river water. Also, any compounds with a nitro group that can be reduced at similar potential as CAP is susceptible to interfere. See for example Zhai et al, 2015, doi: 10.1016/j.electacta.2015.03.140

Reply: Thank you for your suggestion. To consider more interfering compounds specific to river water, one category of reducing substance as ascorbic acid and one antibiotic as malachite green have been added into the interference study. The anti-interference of CNTs/AgNWs electrodes are shown in Figure 5b. One antibiotic with a nitro group named as Nitrofurazone was applied into the interference study. As shown in Figure R, the reduction peak of Nitrofurazone is occurred at reduction potential of -0.35 V, which shows potential discrepancy of nearly -0.15V. The result indicates that Nitrofurazone is not susceptible to interfere CAP detection, and the two antibiotic CAP and Nitrofurazone can be simultaneously electrochemical detected by DPV voltammetry, which need to be further researched. 

 Figure R2. (a) Good selectivity of our electrodes against interferences. (b) Photographs of CAP and Nitrofurazone. (c) Interference study of CAP detection in the presence of Nitrofurazone.

  1. The accuracy of the sensor should be investigated with a set of spiked river samples, if indeed river water is the target sample matrix. The spiking procedure and any dilution of the spiked sample should be clearly explained. The spiking amounts should be consistent with any maximum residue limits or with typical concentrations of CAP reported for river water.

Reply: Thank you for your suggestion. In order to evaluate the practical application performance of CNTs/AgNWs electrodes as CAP sensors, recovery experiments have been performed for the determination of certain concentrations of CAP in real water samples, as shown in Table R1 and Figure R3. The specific procedure for samples preparation have been included in the manuscript. 1 μM and 10 μM CAP have been chosen as the typical concentrations to be added in the river water. A description about this conclusion has been added in the manuscript in Section 3.4 (Line 14 – 20, Page 9).

Figure R3. DPV curves of CNTs/AgNWs electrodes for real sample analysis.

Table R1. Recovery results of CAP in real water samples by using CNTs/AgNWs electrodes.

Samples

Added (μM)

Founded (μM)

RSD (%)

Recovery (%)

Tap water

1

1.073

4.58

107.3

10

9.77

4.47

97.7

River water

1

1.097

4.76

109.7

10

10.700

3.55

107.0

  1. Lines 250-252: the conclusion is contradicted by the data in figure 5c: “Apparently, as shown in Figure 5c, no obvious differences in the potential and current of the characteristic peak were existed between these two samples, indicating that the proposed CNTs/AgNWs electrodes exhibited a good potential for practical CAP determination”. Actually, in figure 5c, the peak height for CAP reduction is quite different in river water than in purified water (“Laboratorial water”) Please investigate the effect of sample matrix and correct the sample preparation to enable accurate measurements.

Reply: Thank you for your suggestion. Taken the effect of real sample matrix to the detection performance into consideration, the specific procedure for samples preparation as pretreatment of real water samples and the spiking amounts of CAP have been conducted to enable accurate measurements. The DPV curves for CAP determination in real sample analysis have been adjusted as in Figure R3.

References:

[1] Mani, V.; Dinesh, B.; Chen, S.M.; Saraswathi, R. Direct electrochemistry of myoglobin at reduced graphene oxide-multiwalled carbon nanotubes-platinum nanoparticles nanocomposite and biosensing towards hydrogen peroxide and nitrite. Biosens. Bioelectron, 2014, 53, 420-427.

[2] Yia, W.W.; Lia, Z.Q.; Dong, C.; Li, H.W.; Li, J.F. Electrochemical detection of chloramphenicol using palladium nanoparticles decorated reduced graphene oxide. Microchem. J, 2019, 148, 774-783.

[3] Yadav, M.; Ganesan, V.; Gupta, R.; Yadav, D.K.; Sonkar, P.K. Cobalt oxide nanocrystals anchored on graphene sheets for electrochemical determination of chloramphenicol. Microchem. J, 2019, 146, 881-887.

[4] Viliana, A.T.E.; Ohb, S.Y.; Rethinasabapathy, M.; Umapathi, R.; Hwang, S.K.; Oh, C.W.; Park, B.; Huh, Y.S.; Han, Y.K. Improved conductivity of flower-like MnWO4 on defect engineered graphitic carbon nitride as an efficient electrocatalyst for ultrasensitive sensing of chloramphenicol. J. Hazard. Mater, 2020, 399, 122868.

We appreciate for Editor/Reviewers’ warm work earnestly, and hope that the correction will meet with approval. The manuscript has been overall checked, and the changes marked in red font one by one. We hope that these revisions are sufficient to make our manuscript acceptable for publication in Sensors. If you have any question about this paper, please do not hesitate to contact me.

Yours sincerely,

Cheng-Te Lin

Ningbo Institute of Material Technology & Engineering, Chinese Academy of Sciences

Round 2

Reviewer 2 Report

1. Storage stability of a material or a composite is very important. Therefore, the authors should provide a brief discussion about the storage stability of CNTs/AgNWs in the main text. Fig. R2a can be moved to the supporting information.

2. The Zeta potential measurement tells us about the surface charge on the individual components, AgNWs, and CNTs and that present on the composite surface leading to Van der Waals interactions. As a result, it is better to discuss this in the first paragraph in section 3.1. where the intertwining effect is discussed. Fig. 3(i) should be moved to Fig. 2(i). 

3. It is well known that two CV segments correspond to one cycle and so 12 CV segments correspond to 6 cycles. Please correct this mistake in section 2.4.

4. Glass carbon electrode (GCE) on page 2 line 46 must be corrected to Glassy carbon electrode (GCE).

In section 2.4, please correct the sentence " Parameters in CV tests.. " as "CV curves (12 cycles) were recorded from -0.8 - 0.4 V with a scan rate of 0.1 V/s."

Author Response

Dear Editor,

We are very grateful to your and the reviewers’ critical comments and thoughtful suggestions. Based on these comments, we have made a careful revision of the original manuscript (Manuscript ID: sensors-1072144). A revised manuscript has been resubmitted, of which the modified sections are marked in red. Thank you and reviewers again, who made great contribution to improve our paper. We responded point by point to the reviewers’ comments as listed below, along with a clear indication of the location in the manuscript:

Reviewer #2:

  1. Storage stability of a material or a composite is very important. Therefore, the authors should provide a brief discussion about the storage stability of CNTs/AgNWs in the main text. Fig. R2a can be moved to the supporting information.

Reply: Thank you for your suggestion. A brief discussion about the storage stability of CNTs/AgNWs has been provided in the main text, as “Considering the long-term stability of electrochemical experiments about CAP detection, the storage stability of CNTs/AgNWs dispersion in ethanol at room temperature is necessary to be investigated. As the results shown in Figure S1a, the CNTs/AgNWs dispersion with 1 : 1 mCNTs/mAgNWs exhibited good storage stability without the appearance of precipitates after standing for 72 h. In order to analyze the degree of long-term stability of the dispersion, the sample was diluted 30 times to increase the transmittance for UV-Vis measurements. In Figure S1b, both the transmittance and absorbance of the dispersion vary slightly with an increase of standing time (12 h), confirming good dispersion stability of CNTs/AgNWs in ethanol at room temperature.”

And Fig.R2 has been moved to the supporting information as Figure S1. This description have been mentioned in the manuscript in Section 3.1 (Line 19-25, Page 4).

  1. The Zeta potential measurement tells us about the surface charge on the individual components, AgNWs, and CNTs and that present on the composite surface leading to Van der Waals interactions. As a result, it is better to discuss this in the first paragraph in section 3.1. where the intertwining effect is discussed. Fig. 3(i) should be moved to Fig. 2(i). 

Reply: Thank you for your suggestion. The discussion about the physical interaction between AgNWs and CNTs with the Zeta potential measurement has been included into the first paragraph in section 3.1, as “To interpret the physical interaction occurred between AgNWs and CNTs, Zeta potential of CNTs, AgNWs and CNTs/AgNWs dispersion was determined to be -24.3 mV, -8.79 mV and -20.0 mV, respectively in Figure 2g. The results indicated the negative surface charges of these samples in ethanol dispersion and suggested Van der Waals interaction between CNTs and AgNWs to form physical-intertwining structure of the composite.”

Fig. 3(i) has been moved to Fig. 2(g). This description has been mentioned in the manuscript in Section 3.1 (Line 15-19, Page 3).

  1. It is well known that two CV segments correspond to one cycle and so 12 CV segments correspond to 6 cycles. Please correct this mistake in section 2.4.

Reply: Thank you for your suggestion. The mistake has been corrected in section 2.4, as “CV curves (6 cycles) were recorded from -0.8 – 0.4 V with scan rate of 0.1 V/s..”. This description have been mentioned in the manuscript in Section 2.4 (Line 7, Page 3).

  1. Glass carbon electrode (GCE) on page 2 line 46 must be corrected to Glassy carbon electrode (GCE).

In section 2.4, please correct the sentence " Parameters in CV tests.. " as "CV curves (12 cycles) were recorded from -0.8 - 0.4 V with a scan rate of 0.1 V/s."

Reply: Thank you for your suggestion. Glass carbon electrode (GCE) has been corrected to Glassy carbon electrode (GCE). The sentence has been corrected as “CV curves (6 cycles) were recorded from -0.8 – 0.4 V with scan rate of 0.1 V/s..”. This description have been mentioned in the manuscript in Section 2.4 (Line 7, Page 3).

We appreciate for Editor/Reviewers’ warm work earnestly, and hope that the correction will meet with approval. The manuscript has been overall checked, and the changes marked in red font one by one. We hope that these revisions are sufficient to make our manuscript acceptable for publication in Sensors. If you have any question about this paper, please do not hesitate to contact me.

Yours sincerely,

Cheng-Te Lin

Ningbo Institute of Material Technology & Engineering, Chinese Academy of Sciences

Reviewer 3 Report

The authors made some of the suggested corrections and changes but the main criticism remains, i.e., the sensor’s selectivity and accuracy for the chosen application, the analysis of tap and river water is not demonstrated. The revised article does not offer an objective representation of the sensor performance: 1) with one exception, the comparison of analytical performances in Table 1 includes only “inferior” sensors (i.e with higher detection limits), excluding the numerous sensors with much better characteristics reported in the literature. 2) The claimed selective detection of CAP in river water is not supported by the data provided on  two (fish pond) water samples, spiked at 1000 higher concentrations than the typical levels of antibiotics in river water. The manuscript is not suitable for publication in Sensors in its current form.

Specific suggestions and comments:

Analysis of river water: as detailed below it is obvious that the linear range and the spiking study do not support the application of the sensor in the analysis of CAP in river water. As a suggestion: the sensor could be appropriate for the determination of nitroaromatic antibiotics in pharmaceutical drugs.

The authors claim in their conclusion” In addition, the CNTs/AgNWs-302 based CAP sensor exhibited good selectivity and behaved suitable in analysis of real water samples, nonetheless the last phrase in the conclusions cautions on the application in real water since other nitro-containing compounds may interfere. The data presented in the answer to reviewer’s comments (voltammogram of mixed CAP and nitrofurazone) should be included in the interference study in the main text and the figure given as well.

The analyzed samples in Table 2 have 1µM and 10 µM CAP. Considering the 100 fold dilution (see line 284), this corresponds to a concentration of CAP in river water of 100 µM and 10 mM, respectively. However literature data indicate typical levels of µg/L of antibiotics in river water. For example, in the study of Liu et al, 2009, https://doi.org/10.1039/B820492F, the maximum concentration of CAP in river water was 47 µg/L, i.e. 0.14 µM. Diluting such a sample 100 fold with PBS would bring the concentration to 1.4 nM, which is less than the detection limit of 0.08 µM of the sensor presented in the manuscript.

The authors should consider in the interference study other nitroaromatic compounds in river water, e.g., nitrobenzene, see Xiao et al, 2015, doi:10.3808/jei.201500323

The voltammograms of  spiked samples including the control sample (unspiked water) should be included in the manuscript, at least as Supplementary information The data should be checked for any mistakes, for example in Table 2 and in the main text are presented results for  river water, while in the R3 figure in the response to reviewer’s comments are shown voltammograms for fish pond water. So, what was anayzed: river water or fish pond water? In any case, the origin of the sample should be given (specift which river or which fish pond).

Comparison of sensor performance with other sensors in literature in Table 1: please include also sensors with better detection limits, e.g.,

Xiao et al, https://doi.org/10.1016/j.talanta.2017.01.078, DL :2.9 nM

Yang et al, 2015, DOI : 10.1016/j.bios.2014.09.041             , DL : 0.1 nM

Kaewnu et al, 2020,  DL: 3 nM, DOI: 10.1149/1945-7111/ab8ce5

Zhai et al, 2015, doi 10.1016/j.electacta.2015.03.140, DL : 0.01 µM

Author Response

Dear Editor,

We are very grateful to your and the reviewers’ critical comments and thoughtful suggestions. Based on these comments, we have made a careful revision of the original manuscript (Manuscript ID: sensors-1072144). A revised manuscript has been resubmitted, of which the modified sections are marked in red. Thank you and reviewers again, who made great contribution to improve our paper. We responded point by point to the reviewers’ comments as listed below, along with a clear indication of the location in the manuscript:

Reviewer #3:

  1. Analysis of river water: as detailed below it is obvious that the linear range and the spiking study do not support the application of the sensor in the analysis of CAP in river water. As a suggestion: the sensor could be appropriate for the determination of nitroaromatic antibiotics in pharmaceutical drugs.

Reply: Thank you for raising this question. Due to a description error about the 100 fold dilution of real samples made in the previous time, but in reality the recovery experiments were conducted without dilution of real samples. So recovery experiments were conducted once again for the determination of certain concentrations of CAP without dilution of real samples. To evaluate recovery performance of the proposed sensor, real samples were then spiked with 0.1 μM and 1 μM CAP. As shown in Figure R2 and Table R1, the results demonstrated that the proposed CNTs/AgNWs electrodes can be appropriate for practical CAP determination in river waters containing typical levels of 0.1μM CAP.

  1. The authors claim in their conclusion” In addition, the CNTs/AgNWs-302 based CAP sensor exhibited good selectivity and behaved suitable in analysis of real water samples, nonetheless the last phrase in the conclusions cautions on the application in real water since other nitro-containing compounds may interfere. The data presented in the answer to reviewer’s comments (voltammogram of mixed CAP and nitrofurazone) should be included in the interference study in the main text and the figure given as well.

Reply: Thank you for your suggestion. The interference study data of CAP detection in the presence of Nitrofurazone has been included in the main text in Figure 6c, and the corresponding DPV curves have been given in Figure S2(a). As shown in Figure R1, the reduction peak potential of Nitrofurazone shifted positively 0.06 V towards CAP, while the peak potential and peak current of CAP remained unchanged. The results indicated that the addition of Nitrofurazone did not interfere CAP detection. This description have been mentioned in the manuscript in Section 3.4 (Line 16-18, Page 9).

Figure R1. Interference study of CAP detection in the presence of Nitrofurazone.

  1. The analyzed samples in Table 2 have 1µM and 10 µM CAP. Considering the 100 fold dilution (see line 284), this corresponds to a concentration of CAP in river water of 100 µM and 10 mM, respectively. However literature data indicate typical levels of µg/L of antibiotics in river water. For example, in the study of Liu et al, 2009, https://doi.org/10.1039/B820492F, the maximum concentration of CAP in river water was 47 µg/L, i.e. 0.14 µM. Diluting such a sample 100 fold with PBS would bring the concentration to 1.4 nM, which is less than the detection limit of 0.08 µM of the sensor presented in the manuscript.

Reply: Thank you for your suggestion. Due to a description error about the 100 fold dilution of real samples made in the previous time, but in reality the recovery experiments were conducted without dilution of real samples. The real water preparation method was described in the manuscript in Section 3.4 (Line 23-25, Page 10). Thank you for pointing out this mistake, we have corrected that. To achieve the practical application of the proposed sensor, real samples were then spiked with 0.1 μM and 1 μM CAP, as indicated in Figure R2. As shown in Table R1, the results demonstrated the accuracy and reliability of the fabricated sensor, indicating that the proposed CNTs/AgNWs electrodes exhibited a good potential for practical CAP determination. This description have been mentioned in the manuscript in Section 3.4 (Line 21-29, Page 10).

Figure R2. DPV curves of CNTs/AgNWs electrodes for real samples analysis.

Table R1. Recovery results of CAP in real water samples by using CNTs/AgNWs electrodes.

Samples

Added (μM)

Founded (μM)

RSD (%)

Recovery (%)

Tap water

0.100

0.109

1.48

109

1.00

0.983

4.55

98.3

River water

0.100

0.111

3.03

111

1.00

1.08

4.88

108

  1. The authors should consider in the interference study other nitroaromatic compounds in river water, e.g., nitrobenzene, see Xiao et al, 2015, doi:10.3808/jei.201500323

Reply: Thank you for your suggestion. As other nitroaromatic compounds as nitrobenzene is not available for us within a short period of time, the antibiotic containing nitro group as Metronidazole was substituted for the interference study. As shown in Figure R3, the reduction peak potential of Metronidazole remained the same with CAP, while the peak current raised largely from 17.8 μA to 83.8 μA when a certain amount of Metronidazole added, indicating the obvious interference of Metronidazole towards CAP. Thus, Metronidazole should be separated before the detection of CAP in real waters containing Metronidazole. This description have been mentioned in the manuscript in Section 3.4 (Line 18-21, Page 9).

Figure R3. Interference study of CAP detection in the presence of Metronidazole.

  1. The voltammograms of spiked samples including the control sample (unspiked water) should be included in the manuscript, at least as Supplementary information The data should be checked for any mistakes, for example in Table 2 and in the main text are presented results for river water, while in the R3 figure in the response to reviewer’s comments are shown voltammograms for fish pond water. So, what was analyzed: river water or fish pond water? In any case, the origin of the sample should be given (specific which river or which fish pond).

Reply: Thank you for your suggestion. The voltammograms of spiked samples including the control sample (unspiked water) has been included in the supplementary information as Figure S3b. The real water samples used in our research is determined to tap water and river water, which has been specified in Figure R2 and Table R1.

  1. Comparison of sensor performance with other sensors in literature in Table 1: please include also sensors with better detection limits, e.g.,

Xiao et al, https://doi.org/10.1016/j.talanta.2017.01.078, DL :2.9 nM

Yang et al, 2015, DOI : 10.1016/j.bios.2014.09.041       , DL : 0.1 nM

Kaewnu et al, 2020,  DOI: 10.1149/1945-7111/ab8ce5,  DL: 3 nM,

Zhai et al, 2015, doi 10.1016/j.electacta.2015.03.140,   DL : 0.01 µM

Reply: Thank you for your suggestion. Sensors with better detection limits in literatures provided have been included in Table R2, as follows.

Table R2. Performance comparison of modified electrodes prepared by various methods for CAP detection.

Modified electrodes

Measurements

Linear range (μM)

LOD (μM)

ref

Chemical synthesis

CNTs/CuNPs

CV

10 – 500

10

[1]

N-doped Graphene/AuNPs

LSV

2.0 – 80

0.59

[2]

rGO/ Co3O4

CV

1.0 – 2000

0.55

[3]

rGO/ ZnO

LSV

0.19 – 2847

0.13

[4]

rGO/ Pt-Pd nanocubes

LSV

0.20 – 30

0.10

[5]

rGO/PdNPs

DPV

0.05 – 1.0

0.05

[6]

Graphene/AgNPs

DPV

0.02 – 20

0.01

[7]

Poly (Eriochrome black T)

SWV

0.01 – 4.0

0.003

[8]

Exfoliated porous carbon

SWV

0.01 – 1.0

0.003

[9]

CNTs/Molecularly imprinted polymer

DPV

0.005 – 4.0

0.0001

[10]

Enzymatic method

Alcohol dehydrogenase

Amperometry

3 – 5000

1

[11]

Physical blending

AuNPs/GO

Amperometry

1.5 – 2.95

0.25

[12]

CNTs/AgNWs

DPV

0.1 – 100

0.08

This work

We appreciate for Editor/Reviewers’ warm work earnestly, and hope that the correction will meet with approval. The manuscript has been overall checked, and the changes marked in red font one by one. We hope that these revisions are sufficient to make our manuscript acceptable for publication in Sensors. If you have any question about this paper, please do not hesitate to contact me.

Yours sincerely,

Cheng-Te Lin

Ningbo Institute of Material Technology & Engineering, Chinese Academy of Sciences

References

[1] Munawar, A.; Tahir, M.A.; Shaheen, A.; Lieberzeit, P.A.; Khan, W.S.; Bajwa, S.Z. Investigating nanohybrid material based on 3D CNTs @ Cu nanoparticle composite and imprinted polymer for highly selective detection of chloramphenicol. J. Hazard. Mater, 2018, 342, 96-106.

[2] Borowiec, J.; Wang, R.; Zhu, L.; Zhang, J.D. Synthesis of nitrogen-doped graphene nanosheets decorated with gold nanoparticles as an improved sensor for electrochemical determination of chloramphenicol. Electrochim. Acta, 2013, 99, 138-144.

[3] Yuan, Q.L.; Liu, Y.; Ye, C.; Sun, H.Y.; Dai, D.; Wei, Q.P.; Lai, G.S.; Wu, T.Z.; Yu, A.M.; Fu, L.; Chee, K.W.A.; Lin, C.-T. Highly stable and regenerative graphene-diamond hybrid electrochemical biosensor for fouling target dopamine detection. Biosens. Bioelectron, 2018, 111, 117-123.   

 [4] Selvi. S.V, Nataraj. N, Chen, S.-M. The electro-catalytic activity of nanosphere strontium doped zinc oxide with rGO layers screen-printed carbon electrode for the sensing of chloramphenicol. Microchem. J, 2020, 159, 105580.

[5] Kong, F.Y.; Luo, Y.; Zhang, J.W.; Wang, J.Y.; Li, W.W.; Wang, W. Facile synthesis of reduced graphene oxide supported Pt-Pd nanocubes with enhanced electrocatalytic activity for chloramphenicol determination. J. Electroanal. Chem, 2016, 781, 389-394.

[6] Schoen, D.T.; Schoen, A.P.; Hu, L.B.; Kim, H.S.; Heilshorn, S.C.; Cui, Y. High Speed Water Sterilization Using One-Dimensional Nanostructures. Nano. Lett, 2010 , 10, 3628-3632. 

[7] Zhai, H.Y.; Liang, Z.X.; Chen, Z.G.; Wang, H.H.; Liu, Z.P.; Su, Z.H.; Zhou, Q. Simultaneous detection of metronidazole and chloramphenicol by differential pulse stripping voltammetry using a silver nanoparticles/sulfonate functionalized graphene modified glassy carbon electrode. Electrochim. Acta, 2015, 171, 105-113.

[8] Kaewnu, K.; Promsuwan, K.; Kanatharana, P.; Thavarungkul, P.; Limbut, W. A simple and sensitive electrochemical sensor for chloramphenicol detection in pharmaceutical samples. J. Electrochem. Soc, 2020, 167, 087506.

[9] Xiao, L.L.; Xu, R.Y.; Yuan, Q.H.; Wang, F. Highly sensitive electrochemical sensor for chloramphenicol based on MOF derived exfoliated porous carbon. Talanta, 2017, 167, 39-43.

[10] Yang, G.M.; Zhao F.Q. Electrochemical sensor for chloramphenicol based on novel multiwalled carbon nanotubes @molecularly imprinted polymer. Biosens. Bioelectron, 2015, 64, 416-422.

[11] Tomassetti, M.; Angeloni, R.; Martini, E.; Castrucci, M.; Campanella, L. Enzymatic DMFC device used for direct analysis of chloramphenicol and a comparison with the competitive immunosensor method. Sensor. Actuat. B-Chem, 2018, 255, 1545-1552.

[12] Karthik, R.; Govindasamy, M.; Chen, S.M.; Mani, V. ; Lou, B.S.; Devasenathipathy, R.; Hou, Y.S.; Elangovan, A. Green synthesized gold nanoparticles decorated graphene oxide for sensitive determination of chloramphenicol in milk, powdered milk, honey and eye drops. J. Colloid. Interf. Sci, 2016, 475, 46-56.

Round 3

Reviewer 3 Report

The authors addressed all issues. The manuscript can be recommended for publication after double checking the spelling throughout the manuscript